# Embodimetrics: A Principal Component Analysis Study of the Combined Assessment of Cardiac, Cognitive and Mobility Parameters

**DOI:** 10.3390/s24061898

**Published:** 2024-03-15

**Authors:** Andrea Chellini, Katia Salmaso, Michele Di Domenico, Nicola Gerbi, Luigi Grillo, Marco Donati, Marco Iosa

**Affiliations:** 1Behaviour & Movement, 50142 Firenze, Italy; kellini72@yahoo.it (A.C.); katiasalmaso@be-move.it (K.S.); 2Embodimetria, 65121 Pescara, Italy; micheledidomenicopt@gmail.com; 3Embodimetria, 56038 Ponsacco, Italy; gerbi.nicola@gmail.com; 4Smart Lab, IRCCS Santa Lucia Foundation, 00179 Rome, Italy; l.grillo@hsantalucia.it; 5Motustech, 00012 Guidonia, Italy; marco.donati@motustech.it; 6Department of Psychology, Sapienza University, 00185 Rome, Italy

**Keywords:** embodiment, heart rate variability, Stroop task, artificial intelligence, motor control

## Abstract

There is a growing body of literature investigating the relationship between the frequency domain analysis of heart rate variability (HRV) and cognitive Stroop task performance. We proposed a combined assessment integrating trunk mobility in 72 healthy women to investigate the relationship between cognitive, cardiac, and motor variables using principal component analysis (PCA). Additionally, we assessed changes in the relationships among these variables after a two-month intervention aimed at improving the perception–action link. At baseline, PCA correctly identified three components: one related to cardiac variables, one to trunk motion, and one to Stroop task performance. After the intervention, only two components were found, with trunk symmetry and range of motion, accuracy, time to complete the Stroop task, and low-frequency heart rate variability aggregated into a single component using PCA. Artificial neural network analysis confirmed the effects of both HRV and motor behavior on cognitive Stroop task performance. This analysis suggested that this protocol was effective in investigating embodied cognition, and we defined this approach as “embodimetrics”.

## 1. Introduction

Any comprehensive model of wellness should account for a complex mix of cognitive, affective, behavioral, and physiological factors that contribute to individual differences in health and disease [1]. These individual differences related to blood and pulse pressures are often associated with autonomic balance and may influence cognitive performance [1]. In fact, there is a growing body of literature that highlights the relationship between vagally mediated heart rate variability (HRV) and good performance in cognitive tasks that require the use of mental functions [1,2]. HRV is a widely used measure due to its convenience and noninvasive features. It is also associated with the activation of the sympathetic nervous system (SNS), resulting in a decrease in its high-frequency components (and a relevant increase in low-frequency components) when the sympathetic nervous system is activated [3,4].

Heart rate variability was also found to be associated with Stroop task performance [5], which is frequently used to assess the ability to manage interfering information at a cognitive level [6]. As part of the Stroop task, the subject is asked to verbally read a word that represents the name of a color, which may be written in the same color semantically represented by that word (congruent condition) or in a different color (incongruent condition). It is a widely used cognitive test that assesses the ability to regulate thoughts and actions in accordance with internally maintained behavioral goals, achieved through the activation of cognitive control mechanisms [7].

In recent decades, there has been an increasing proliferation of wearable devices capable of measuring cardiac functions, electrodermal activity, skin temperature, electromyographic activity, and kinematic parameters of trunk movements, thanks to embedded inertial sensors [8,9]. More recently, researchers have attempted to utilize these wearable devices to investigate the complex relationships between cardiac functions, cognitive aspects, and movement control [10]. Indeed, movement control necessitates the integration of sensorial feedback with an internal body representation and often involves higher-order cognitive processes. For instance, it has been observed that the faster individuals walk, the more closely their cardiac rhythm is coupled with cognitive performance [10]. 

According to the concept of a comprehensive model, the relationship between physiological factors related to the autonomic system and cognitive performance should also consider motor and behavioral aspects. Despite the growing body of literature on this topic, there have been no protocols for testing whether motor control may play a role in the coupling between HRV and Stroop task performance. However, there are numerous direct and indirect pathways linking the frontal cortex to autonomic motor circuits responsible for both the sympatho-excitatory and parasympatho-inhibitory effects on the heart [1]. On the other hand, a wide range of inhibitory processes across cognitive, motor, and affective tasks are associated with the same brain region, known as the right prefrontal cortex [1]. This may explain why motor awareness can be reduced in the presence of cognitive load [11].

Previous studies have investigated the relationship between cognitive load and heart rate variability [5,12], spanning from analyses of this relationship in individuals with neurodegenerative diseases [13] to those with a high level of physical fitness [14].

Studying the complex system that includes cognitive, motor, and cardiac functions is essential for identifying the principal components that are transversally involved in all three systems.

The aim of this study was to investigate, using a simple protocol, the relationship between the heart, motor control, and cognitive functions by analyzing the principal components of this complex system. Based on the existing literature, the protocol focused on analyzing HRV and its relationship to Stroop task results, while also incorporating the analysis of trunk rotations. It has been reported that an increase in trunk mobility may alter the sympathovagal balance, thereby modifying HRV [15]. Trunk rotations were measured using a wearable inertial unit containing a triaxial accelerometer, a triaxial gyroscope, and a magnetometer (used for measuring the range of motion) [8,9]. From a bioengineering perspective, a device embedding inertial sensors for analyzing trunk movements and electrodes for recording cardiac signals to compute heart rate variability was proposed. Subsequently, we tested whether this protocol utilizing sensors could sensitively detect changes induced by a specific physical intervention aimed at enhancing the perception–action link, which is fundamental to embodied cognition [16]. Given previous findings indicating gender differences in autonomic cardiac control [17], trunk accelerations [18], and Stroop task performance [19], we enrolled only women in this study to simplify the variables to be controlled. Furthermore, due to the recent emergence of artificial neural networks as a frontier in data analysis [20], including those used for assessing heart rate variability [21,22], we utilized an artificial neural network (ANN) to verify the relationship between principal components and cognitive outcomes.

## 2. Materials and Methods

### 2.1. Participants

We enrolled a sample of 72 healthy women in this study (mean age: 49.4 ± 7.7). All of them were free from orthopedic, neurological, or psychological diseases. Signed informed consent was obtained from all participants prior to their involvement in the study. 

### 2.2. Protocol

Participants underwent two assessments. They sat on a comfortable table wearing a sensorized trunk band (Beyond Inertial, Motustech, Rome, Italy), as depicted in Figure 1. The device embedded an inertial measurement unit with a triaxial accelerometer, a triaxial gyroscope, and a magnetometer, as well as electrodes for assessing the heart signal. The assessment protocol involved measuring heart rate variability at baseline for 5 min. Subsequently, participants were asked to perform trunk rotations with their arms flexed on the trunk on the left and right side. 

The device enabled the measurement of intervals between successive R waves (RR intervals). From these data, heart rate variability (HRV) was assessed by examining the temporal variations within these intervals. HRV analysis was conducted in both the time and frequency domains. In the frequency domain, the analysis involved decomposing the signal into distinct frequency components. The designated frequency bands included high frequencies (HRV-HF: 0.15–0.4 Hz), low frequencies (HRV-LF: 0.04–0.15 Hz), and very low frequencies (HRV-VLF: 0.01–0.04 Hz). The percentage of the signal in each one of these frequency domains was analyzed according to the literature, indicating that HF is mediated by the vagal nerve of the parasympathetic nervous system (PNS), whereas LF mainly reflects the activity of the sympathetic nervous system. The balance between these two systems is reflected in their respective contributions [23]. 

The inertial measurement unit (IMU) was utilized to quantify accelerations, angular velocities, and magnetic field orientations within a given reference system. The acquisition of these measurements served as a precursor for subsequent analyses. By employing a sensor fusion algorithm, the IMU enables the estimation of the sensor’s orientation within the inertial reference system [24]. The algorithmic integration facilitated the determination of rotation angles corresponding to left and right rotations. We computed the trunk range of motion (ROM) as the sum of left and right rotations, taking the absolute values into account. Additionally, the symmetry index (SI) was calculated as the ratio between the lower and the higher angle, multiplied by 100, resulting in a value of 100% for two equal (symmetric) rotations of the trunk in both direction [25]. Then, participants were asked to perform a classical version of the Stroop task with 15 words, and the time to complete the task (TCT) was recorded along with the number of errors made (n_e_). From these two parameters, we computed the percentage accuracy (ACC) of the subject using the formula (15 − n_e_)/15 × 100 and the normalized time to complete the Stroop task (NTCT) was calculated by combining time and n_e_ as follows: TTC × 15/(15 − n_e_). After the initial assessment, subjects were re-assessed following a two-month intervention. During this intervention, participants received a pamphlet outlining a protocol to follow. This protocol required them to engage in outdoor walking for at least 20 min per day whilst breathing through the nose. Additionally, participants were instructed to improve their perception and body control by moving their bare feet on a soft small ball while maintaining an upright posture maintained with slightly flexed knees for 5 min per day. They alternated between using both feet and paid attention to the perception–action link during these activities. All subjects verbally reported the adherence to the protocol. 

### 2.3. Statistical Analysis

The data were reported in terms of means and standard deviations. Pearson’s coefficient (R) was utilized to assess the correlation between variables, with the relevant regression equations as y = a × x + b. The main analysis of this study aimed to identify the basic components among the three assessments (cardiac variables: HRV-VLF, HRV-LF, and HRV-HF; motor variables: ROM and SI; and cognitive variables: ACC and NTCT) was principal component analysis (PCA). This analysis was performed on the variables assessed at baseline and after the intervention to determine how the data could be aggregated. The number of the components was not predetermined, but was based on a parallel analysis conducted on the scree plot. The varimax method was employed as the rotational algorithm. Paired *t*-tests were conducted to compare the measured variables before and after the intervention.

The percentage changes between after and before intervention were computed as the difference between the post- and pre-intervention score, divided by the pre-intervention score and multiplied by 100.

The effects of changes in HRV and the effects of changes in embodied motor behavior on cognitive performance were evaluated using ARIANNA, an artificial intelligent assistant for neural network analysis. ARIANNA was used to predict modifications in cognitive performance in the Stroop test after intervention (output layer). ARIANNA is a multilayer perceptron, formed by the input layer and 2 hidden layers (with 5 nodes in each one) [19]. The architecture of ARIANNA was that of a feed-forward neural network (FFNN), with data moving in only one direction, from the input nodes through the two hidden layers to the output nodes. The activation function for all the units in the hidden layers and for the output layer was a hyperbolic tangent. The chosen computational procedure was based on online training. 

The input layers (corresponding to independent variables) were as follows: trunk ROM, trunk symmetry index, HRV-VLF, HRV-LF, and HRV-HF, whereas the two output layers (dependent variables) were predicted to be Stroop accuracy and Stroop NTCT. 

The alpha level of statistical significance was set at 5% for rejecting the null hypothesis for all the performed analyses. The pieces of software used for all the above analyses were Jamovi (version 2.3.21) and IBM SPSS Statistics (version 23).

## 3. Results

### 3.1. Baseline to Post Intervention Parameters

Table 1 shows the parameters measured at baseline and post-intervention. Firstly, it is notable that HRV-LF did not significantly change, whereas the change in HRV-HF was significant and mainly related to a variation in HRV-VLF, although this modification also did not achieve the threshold for rejecting the null hypothesis. To delve deeper into this aspect, we analyzed HF with respect to LF, as depicted in Figure 1. Pre-intervention, the regression equation was HRV-HF = −0.64 × HRV-LF + 62, with a Pearson correlation coefficient of R = −0.43 (*p* < 0.001). After intervention, this relationship remained statistically significant, but weaker (R = −0.32 (*p* = 0.006)), with a regression equation of HRV-HF = −0.43 × HRV-LF + 51. 

Cognitive performance improved both in terms of accuracy and normalized time to complete the task. From a motor point of view, trunk ROM significantly improved by about 17 degrees, whereas the symmetry index did not change.

### 3.2. Principal Component Analysis

The baseline PCA correctly identified three main domains (Table 2): the first component, mainly contributed to by Stroop task variables (ACC and NTCT); the second component comprising the three frequency domains of HRV; and the third component, consisting of trunk kinematic variables (ROM and SI). Despite this clear subdivision, HRV-LF emerged as a transversal parameter, contributing to all three components.

The protocol and subsequent PCA successfully assessed the changes after the intervention, leading to a re-arrangement of the parameters. As shown in Table 3, only two components were identified, one for cognitive and one for motor control, and another one related to cardiac parameters (HRV-HF and HRV-VLF). HRV-LF was associated with cognitive and motor control, aligning with the aim of intervention based on these three systems.

### 3.3. Artificial Neural Network Analysis

Figure 2 illustrates the architecture of the artificial neural network ARIANNA. The computed importance assigned to each of the input layers is reported in Table 4. The change in Stroop performance was mainly predicted by changes in HRV-LF (21.4%) and those in trunk movement parameters (ΔROM (23.3%) and ΔSI (19.6%)). The accuracy of the ANN in making predictions was high, as shown in Figure 3, for both outputs. 

## 4. Discussion

A sensor-based quantitative assessment of heart rate variability has often been linked to cognitive performance, particularly in tasks like the Stroop task (e.g., in athletes [26], individuals with post-traumatic stress syndrome [27], and healthy subjects managing daily stress [28]), as well as to motor control (including measures such as range of motion and interoceptive accuracy [29]). However, a protocol considering these three domains together was still lacking. Therefore, the primary aim of this study was to establish a simple protocol for identifying the components integrating heart rate variability, trunk motion variability, and cognitive parameters obtained from the Stroop task.

The protocol proposed in our study utilized a sensorized band worn at the trunk level, measuring the temporal distance between R waves and left and right trunk rotations using an embedded inertial device. This protocol included PCA that accurately identified three components related to the aforementioned domains: cardio, cognition, and motion. As a form of confirmatory analysis, we employed an approach based on assessing proficiency in predicting changes in cognitive performance using an artificial neural network, a method previously utilized in the literature [20]. 

According to previous literature studies [30], the balance between HRV-LF and HRV-HF was found to be not only associated with cardiac parameters but also cognitive ones [5], and, as we suggested, also with kinematic parameters. In healthy subjects, the ability to perform wide symmetric rotations not only depends on kinematic functions but also the perception of the trunk mid-line [25], afferent feedback, and the perception–action link [16].

The second aim was to verify if and how it was possible to measure the alteration in these relationships after a specific intervention focused on the embodiment of sensations and cognition. After this intervention, only two components were identified (one related to high and very low frequencies of heart rate variability, and another one combining HRV-LF, parameters related to Stroop task performance, and those related to the execution of wide symmetric trunk rotations. 

This result is not surprising. After many years of cognition and motor control being investigated separately, with a distinct separation between mind and body, numerous authors have began to criticize this perspective. Damasio talked about Descartes’ error of dividing mind and body [31]; Clark argued for the necessity of putting brain, body, and world together again in neuroscience [32]; and Berthoz defined the so-called brain’s sense of movement [33]. 

More recently, the 4E theory of cognition has proposed that the body is a constituent of the mind, and cognition is closely related to physiological parameters and performed motor actions [34]. The theory of embodied cognition and, further, the “4E” approach suggested that cognition does not solely occur in the brain, but is also embodied, embedded, enacted, or extended through body structures, functions, and processes [34,35]. 

According to these recent theories, a recent research reported that the cognitive performance of executive function tasks, which evoke attentional control, partially depends on the responsiveness of autonomic control parameters that can be assessed by heart rate variability [30].

The results of our PCA were further confirmed by the analysis conducted using artificial neural networks to assess the effect of changes in HRV and trunk movements on changes in Stroop task performance. We employed an artificial intelligence tool that has been utilized in various studies [20,36]. The input parameters associated with a higher relative importance were changes in the trunk range of motion, changes in trunk movement symmetry, and changes in HRV-LF. These results corroborated those found by PCA and demonstrated the high predictability of observed outcomes (as shown in Figure 3). Furthermore, artificial neural networks represent an emerging approach that can be used to identify more complex relationship among variables, especially when they are not simply linear, as in this case. For these purposes, the artificial neural network analyses appeared particularly suitable for investigating the parameters extracted by inertial magnetic units, such as those used in this study, as has been carried out in clinical contexts previously [36]. 

De Bartolo and colleagues have advocated for the development of evaluation methods for the quantitative assessment of cardiac, cognitive, and motor interactions, which could be beneficial in physiological research, athlete training, and specifically for rehabilitation purposes [10]. Indeed, neurorehabilitation could benefit from an integrated approach that does not solely aim for the separate recovery of specific functions, but rather focuses on the holistic care and treatment of the patient as a whole person. 

The utilization of a multimodal assessment based on principal component analysis on heart rate variability and brain data is nothing new [37,38]. However, in this study, we proposed a combined approach to quantify the level of overlapping of cognitive, motor, and cardiac functions. We could define this approach as “embodimetrics” because they focused on the assessment of embodiment, akin to how psychometrics is the discipline concerning the quantitative measurement of psychological aspects [39].

The results of this study should be read in light of its limits. Only female participants were involved, adherence to the intervention was verbally reported by participants but not quantitatively assessed, and environmental factors were not taken into account despite evidence suggesting they may influence from cognitive and motor coupling strategies [40,41,42,43]. Then, some variables were only qualitatively assessed. Further research on this topic should be conducted in controlled environments via the quantitative monitoring of participants’ adherence to the protocol, for example using smartwatches to record the frequency and intensity of physical activity. Additionally, future studies should include participants of both genders and incorporate other tests assessing cognitive functions. Furthermore, while artificial neural network analysis provided good results in terms of accuracy, it often lacks high reliability [36]. However, the trustworthiness of our findings was bolstered by principal component analysis, which does not suffer from these reliability issues and yielded similar findings to the ANN. 

In conclusion, our study utilizing a combined assessment of HRV and trunk mobility identified the level of connection with cognitive aspects measured by the Stroop task, providing a useful approach for measuring the strength of the relationship between cognitive functions, autonomic cardiac functions, and body movements. It is important to emphasize the necessity of collecting and analyzing data using a scientific, validated, and reliable approach. The study of embodiment is a new interesting field of psychometrics [44,45,46,47]. Furthermore, in line with the emerging literature on embodied cognition [16] and the theory of 4E [34], we proposed the introduction of a new term to define the field of research within psychometrics concerning the techniques and properties of objective measurements, assessments, and analyses related to the latent construct of embodied cognition and all the aspects related to the 4E theory that we named embodimetrics.

## Figures and Tables

**Figure 1 sensors-24-01898-f001:**
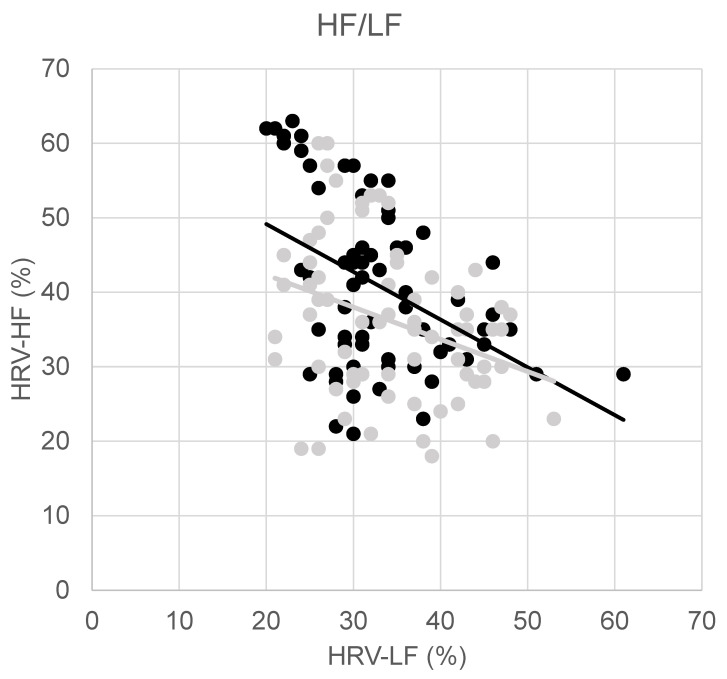
Percentages of heart rate variability (HRV) in low frequencies (LF, x-axis) and high frequencies (HF, y-axis); before (black dots) and after (grey dots) intervention are also shown with the relevant regression lines.

**Figure 2 sensors-24-01898-f002:**
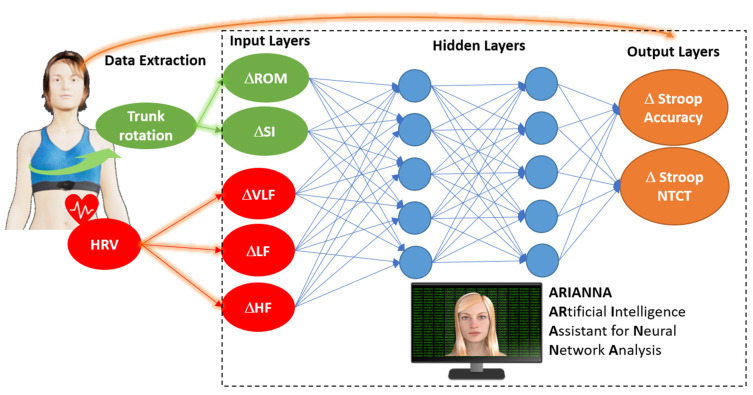
Architecture of artificial intelligence assistance for neural network analysis (ARIANNA). Δ: change in parameters post vs. pre, HRV: heart rate variability, VLF: very low frequency, LF: low frequency, HF: high frequency, ROM: range of motion of the trunk, SI: symmetry index, NTCT: normalized time to complete the Stroop task.

**Figure 3 sensors-24-01898-f003:**
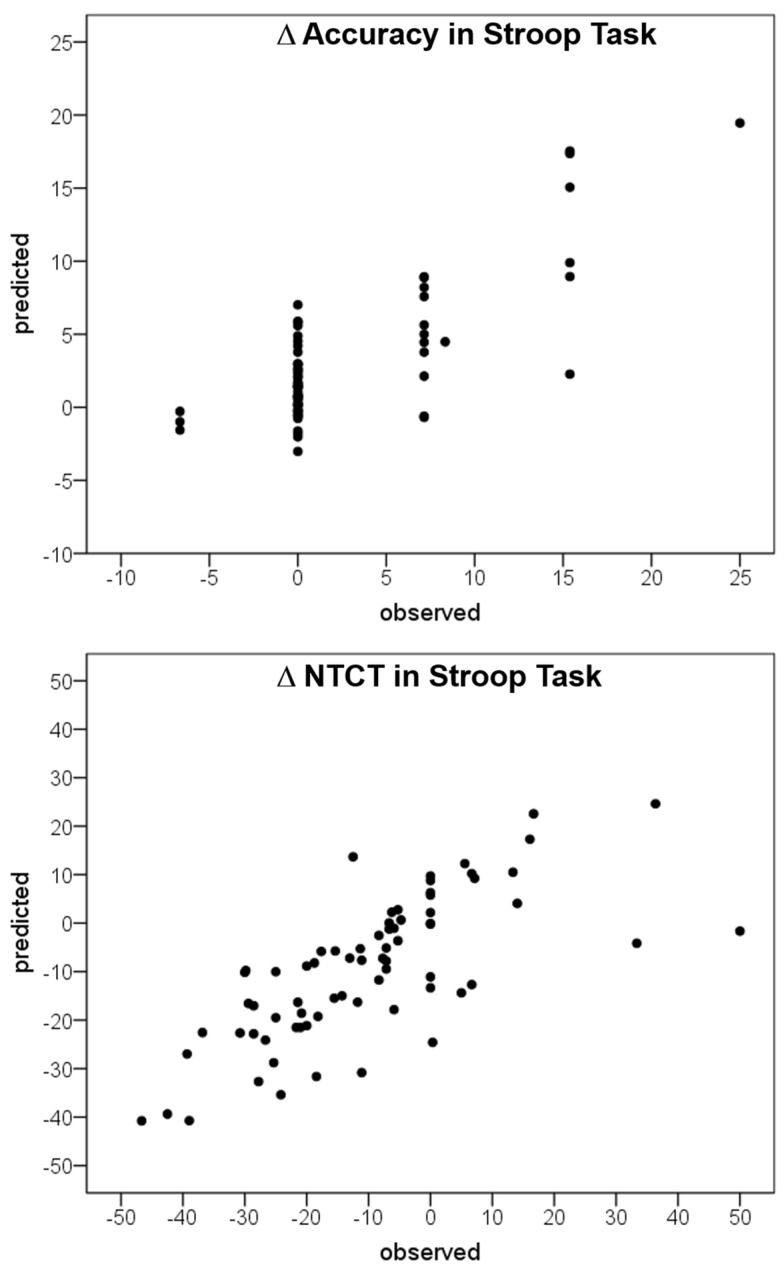
Predictions of changes in performance (Δ) at the Stroop task on the basis of parameters reported in Table 4. NTCT: Normalized time to complete the task.

**Table 1 sensors-24-01898-t001:** Mean ± standard deviation of measured variables pre- and post-intervention with the *p*-value obtained by Wilcoxon rank test (in bold if *p* < 0.05). HRV: heart rate variability, VLF: very low frequency, LF: low frequency, HF: high frequency, ACC: Stroop task accuracy, NTCT: normalized time to complete the Stroop task, ROM: range of motion, SI: symmetry index.

Variable	Pre	Post	*p*-Value
HRV-VLF (%)	26.3 ± 10.2	29.2 ± 10.7	0.096
HRV-LF (%)	33.1 ± 7.5	34.2 ± 7.9	0.522
HRV-HF (%)	40.8 ± 11.1	36.2 ± 10.5	**0.002**
HF/LF	1.3 ± 0.6	1.0 ± 0.5	**0.001**
Stroop task NTCT (s)	18.8 ± 5.2	16.4 ± 3.9	**<0.001**
Stroop task ACC	96.6 ± 5.2	98.9 ± 2.7	**<0.001**
Trunk ROM (deg)	110.0 ± 21.5	127.0 ± 28.0	**<0.001**
Trunk SI (%)	87.4 ± 9.1	87.9 ± 10.0	0.743

**Table 2 sensors-24-01898-t002:** The effects of variables assessed at baseline (pre-intervention) on the components obtained with principal component analysis (in bold if their absolute value is >0.25).

Variable	Component 1	Component 2	Component 3
HRV-VLF	−0.20	**0.84**	−0.19
HRV-LF	**0.34**	**0.36**	**0.43**
HRV-HF	−0.03	**−0.99**	−0.06
Stroop task NTCT	**0.87**	−0.11	−0.07
Stroop task ACC	**−0.83**	0.04	−0.02
Trunk ROM	0.12	−0.09	**0.72**
Trunk SI	−0.13	−0.02	**0.77**

**Table 3 sensors-24-01898-t003:** The effects of variables assessed post-intervention on the components obtained with principal component analysis (in bold if their absolute value is >0.25).

Variable	Component 1	Component 2
HRV-VLF	−0.24	**0.92**
HRV-LF	**0.62**	−0.01
HRV-HF	−0.19	**−0.94**
Stroop task NTCT	**0.80**	0.03
Stroop task ACC	**−0.53**	0.05
Trunk ROM	**−0.27**	0.14
Trunk SI	**0.35**	0.04

**Table 4 sensors-24-01898-t004:** Results of artificial neural network analysis in predicting the normalized time to complete the Stroop task and determine accuracy. Δ: change in the parameters post vs. pre, HRV: heart rate variability, VLF: very low frequency, LF: low frequency, HF: high frequency, ROM: range of motion of the trunk, SI: symmetry index, NTCT: normalized time to complete the Stroop task.

Input Layer Parameters	Importance of the Input Layer in Output Prediction		
	Raw Weight	Relative	Normalized
ΔHRV-VLF	0.191	19.1%	81.9%
ΔHRV-LF	0.214	21.4%	91.8%
ΔHRV-HF	0.166	16.6%	71.4%
ΔTrunk ROM	0.233	23.3%	100%
ΔTrunk SI	0.196	19.6%	84.4%

## Data Availability

Data are available on request from the corresponding author.

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
