# Peer review of "Embodimetrics: A Principal Component Analysis Study of the Combined Assessment of Cardiac, Cognitive and Mobility Parameters"

_sensors, 2024, doi:10.3390/s24061898_

Round 1
Reviewer 1 Report
Comments and Suggestions for Authors
The present study by A. Chellini et al focuses on the combined assessment of HRV and trunk mobility relationship with the cognitive aspects measured by the Stroop task. While the author demonstrates an interesting approach to determining the relationship between HRV and cognitive performance which can be quite useful for the scientific community working in a similar field, there are a few major drawbacks in this study that need to be addressed or acknowledged before publication. After adequate work and the author’s response, these can be reconsidered. Please see below the reviewer’s comments which can help strengthen the manuscript.
· The authors describe a quantitative way of measuring the relationship between HRV and cognitive performances. However, rightfully so, they have mentioned the limitations of their study - adherence to the intervention was verbally reported by participants and others. In my opinion, the authors should acknowledge that their study, at present, is semi-qualitative. More control experiments in controlled environments should be done to substantiate the claim.
· In conclusion, authors have claimed that their study helps measurement of how strong the relationship between cognitive functions and HRV is. However, ANN often lack reliability. In that case, how can this study be used for quantitative measurements? Reviewers think that the conclusion and core message of the manuscript should be modified to reflect this.
· Affiliation of author(s) is incomplete. There is no information on authors superscripted with numbers 4, 5, and 6.
· Line 32 and in the abstract: The use of the word ‘some’ should be avoided in scientific articles.
· Line 36: The abbreviation of ‘SNS’ should be described at its first mention in the manuscript (sympathetic nervous system).
· Line 67-68: The reviewer thinks that the sentence is inaccurate and unfounded. There are multiple studies where researchers have demonstrated the relationship between HRC and cognitive performance. Authors should phrase this sentence differently or acknowledge already published peer-reviewed papers. https://journals.plos.org/plosone/article?id=10.1371/journal.pone.0056935; https://www.sciencedirect.com/science/article/pii/S1568163721002865?via%3Dihub.
· Line 72-87: Reviewer thinks that the message of sentences in this paragraph is not entirely clear. The author should consider rewriting it. Especially 2nd sentence of the paragraph.
· Line 86: The abbreviation of ANN should be mentioned.
· Authors should note a general rule on abbreviations - Abbreviations should only be used if the term appears two or more times in the text. Spell out the full term at its first mention, indicate its abbreviation in parenthesis, and use the abbreviation from then on. There are many instances where this was not followed, authors should check on that.
· Granted there are differences in trunk accelerations depending on gender. Authors should mention in a few words why they focused their analysis only on females.
Comments on the Quality of English LanguageOne additional comment that I would like to mention - There are many instances in the manuscript where English is incomprehensible, at least for the reviewer, and it was reasonably difficult to understand what authors wanted to describe as their statements can be interpreted in more than one way. The reviewer suggests to use smaller sentences
Author Response
AUTHORS: We would like to thank the Editor and the Reviewers for their general positive comments about our work, for the possibility to resubmit it and for the qualified comments that helped us into improving our manuscript in its revised version.
In the following our step-by-step answers to the comments of the Reviewer 1. We have also reported here the new/changed parts of the revised manuscript between apices.
Reviewer 1
The present study by A. Chellini et al focuses on the combined assessment of HRV and trunk mobility relationship with the cognitive aspects measured by the Stroop task. While the author demonstrates an interesting approach to determining the relationship between HRV and cognitive performance which can be quite useful for the scientific community working in a similar field, there are a few major drawbacks in this study that need to be addressed or acknowledged before publication. After adequate work and the author’s response, these can be reconsidered. Please see below the reviewer’s comments which can help strengthen the manuscript.
AUTHORS: Thank you for the general positive comments about our work, especially in relationship to the combined assessment of HRV and Stroop task. We would also thank the reviewer for the following qualified comments that actually helped us into improving our manuscript in its revised version
- R1: The authors describe a quantitative way of measuring the relationship between HRV and cognitive performances. However, rightfully so, they have mentioned the limitations of their study - adherence to the intervention was verbally reported by participants and others. In my opinion, the authors should acknowledge that their study, at present, is semi-qualitative. More control experiments in controlled environments should be done to substantiate the claim.
AUTHORS: The referee is right, so we have added the following sentence about further studies in the paragraph about the limits of our study:
“Then, some variables were only qualitatively assessed: further researchers on this topic should be conducted in controlled environments with quantitative monitoring of par-ticipants’ adherence to the protocol, for example using smartwatches to record the frequency and intensity of physical activities.”
- R1: In conclusion, authors have claimed that their study helps measurement of how strong the relationship between cognitive functions and HRV is. However, ANN often lack reliability. In that case, how can this study be used for quantitative measurements? Reviewers think that the conclusion and core message of the manuscript should be modified to reflect this.
AUTHORS: To clarify this important point after this sentence about the low reliability of ANNs:
“Furthermore, while artificial neural network analysis provided good results in terms of accuracy, it often lacks high reliability [37].”
We have added this one:
“However, the trustworthiness of our find-ings was bolstered by the main Principal Component Analysis, which does not suffer from these reliability issues and yielded similar findings to the ANN.”
Finally, in the core message of our conclusions we have also added the following sentence:
“It is important to emphasize the necessity of collecting and analyzing data using a sci-entific, validated and reliable approach.”
R1: Affiliation of author(s) is incomplete. There is no information on authors superscripted with numbers 4, 5, and 6.
AUTHORS: Sorry, there was some missing information, we have corrected them.
- R1:Line 32 and in the abstract: The use of the word ‘some’ should be avoided in scientific articles.
AUTHORS: We have corrected the two sentences as follows:
ABSTACT: “There is a growing body of literature investigating the relationship between frequency domain analysis of heart rate variability (HRV) and cognitive performance in the Stroop task.”
INTRODUCTION: “In fact, there is a growing body of literature that highlights the relationship between vagally mediated heart-rate variability (HRV) and good performance in cognitive tasks that require the use of mental functions [1,2].”
- R1:Line 36: The abbreviation of ‘SNS’ should be described at its first mention in the manuscript (sympathetic nervous system).
AUTHORS: Thank you, we have added the extended version “sympathetic nervous system (SNS)”
- R1:Line 67-68: The reviewer thinks that the sentence is inaccurate and unfounded. There are multiple studies where researchers have demonstrated the relationship between HRC and cognitive performance. Authors should phrase this sentence differently or acknowledge already published peer-reviewed papers. https://journals.plos.org/plosone/article?id=10.1371/journal.pone.0056935; https://www.sciencedirect.com/science/article/pii/S1568163721002865?via%3Dihub.
AUTHORS: Thanks for this correction. We have completely changed the paragraph as following, adding the two new references (new 12 and 13) suggested by the reviewer:
“Previous studies have investigated the relationship between cognitive load and heart rate variability [5,12], spanning from analyses of this relationship in individuals with neurodegenerative diseases [13] to those with a high level of physical fitness [14].”
- Luque-Casado, A., Zabala, M., Morales, E., Mateo-March, M., & Sanabria, D. Cognitive performance and heart rate variability: the influence of fitness level. PloS one, 2013, 8(2), e56935.
- Liu, K. Y., Elliott, T., Knowles, M., & Howard, R. Heart rate variability in relation to cognition and behavior in neurodegenerative diseases: A systematic review and meta-analysis. Ageing research reviews, 2022, 73, 101539.
R1: Line 72-87: Reviewer thinks that the message of sentences in this paragraph is not entirely clear. The author should consider rewriting it. Especially 2nd sentence of the paragraph.
AUTHORS: We have deeply revised these paragraphs as follows:
“The aim of this study is to investigate, using a simple protocol, the relationship among heart, motor control and cognitive functions by analyzing the principal com-ponents of this complex system. Based on the existing literature, the protocol focused on analyzing HRV and its relationship to Stroop task results, while also incorporating the analysis of trunk rotations. It has been reported that an increase in trunk mobility may alter the sympathovagal balance, thereby modifying HRV [15]. Trunk rotations were measured using a wearable inertial unit containing a triaxial accelerometer, a triaxial gyroscope and a magnetometer for measuring the range of motion [8,9]. From a bioengineering perspective, a device embedding inertial sensors for analyzing trunk movements and electrodes for recording the cardiac signal to compute heart rate variability has been proposed. Subsequently, we tested whether this protocol could sensitively detect changes induced by a specific physical intervention aimed at enhancing the perception-action link, which is fundamental to embodied cognition [16]. Given previous findings indicating gender differences in autonomic cardiac control [17], trunk accelerations [18], and Stroop task performance [19] we enrolled only women in this study to simplify the variables to be controlled. Furthermore, due to the recent emergence of artificial neural networks as a frontier in data analysis [20], including that of heart rate variability [21,22], we utilized an artificial neural network (ANN) to verify the relationship between principal components and the cognitive outcomes.”
- R1:Line 86: The abbreviation of ANN should be mentioned.
AUTHORS: Done writing “we used an artificial neural network (ANN)”
- R1:Authors should note a general rule on abbreviations - Abbreviations should only be used if the term appears two or more times in the text. Spell out the full term at its first mention, indicate its abbreviation in parenthesis, and use the abbreviation from then on. There are many instances where this was not followed, authors should check on that.
AUTHORS: Thank you, we have now corrected it throughout the whole manuscript
- R1:Granted there are differences in trunk accelerations depending on gender. Authors should mention in a few words why they focused their analysis only on females.
AUTHORS: According to this comment we have now specified the reasons behind this choice in the following sentence in which we cited 3 important references:
“Given previous findings indicating gender differences in autonomic cardiac control [17], trunk accelerations [18], and Stroop task performance [19] we enrolled only women in this study to simplify the variables to be controlled.”
- Botek, M., Krejčí, J., & McKune, A. Sex Differences in Autonomic Cardiac Control and Oxygen Saturation Re-sponse to Short-Term Normobaric Hypoxia and Following Recovery: Effect of Aerobic Fitness. Frontiers in endo-crinology, 2018, 9, 697.
- Mazzà, C.; Iosa, M.; Picerno, P.; Cappozzo, A. Gender differences in the control of the upper body accelerations during level walking. Gait Posture, 2009, 29(2), 300-303.
- Mekarski, J.E.; Cutmore, T.R.; Suboski, W. Gender differences during processing of the Stroop task. Percept Mot Skills. 1996, 83(2), 563-8.
R1: Comments on the Quality of English Language - One additional comment that I would like to mention - There are many instances in the manuscript where English is incomprehensible, at least for the reviewer, and it was reasonably difficult to understand what authors wanted to describe as their statements can be interpreted in more than one way. The reviewer suggests to use smaller sentences
AUTHORS: We have performed an English Editing throughout the whole revised manuscript

Reviewer 2 Report
Comments and Suggestions for Authors
General comment:
In line with the 4E cognition theory, the study is carried out with the attempt to apply a new integrative approach called "embodimetrics" to the study of cognitive constructs. Here the performance at the stroop task - one of the literature's gold standard for attentional measures - was measured combining kinestetic and physiological variables. The originality of the study relied on the inclusion of artificial neural networks analyses to further confirm the predictability results from the statistical analyses. The use of a multimodal approach, both theoretical and technological, helps optimize and improve the study of psychological complex phenomena.
Main revisions:
1. The gender differences reported in the previous literature brought the authors to focus on a full women sample. The reason behind this choice (if there is any) is not clear or reported.
2. For the second phase of the assessment, participants had to follow a protocol consisting of activities such as walking and balance exercises. The study reports a general adherence from participants which is vague and lacks control measures. Devices such as smart-watches or fitness-watches could be given to participants as to better track their activities, and most importantly provide quantitative measures to statistically analyse. Overall, this would confer more reliability to this second phase of the experimental protocol.
3.The outline of the independent and dependent variables used in the study are not enough clear, as well as the hypotheses. This may be improved by creating sub-sections in the protocol paragraph. In a first section the variables could be described in their abbreviations and functioning, and separately in another section they could be explained in their expectations/hypotheses.
4. The choice to use the sensory trunk band device "Beyond Inertial" is incomplete to the aim of this study. Is it justified by the advantage of combining both the collection of both heart-physiological and trunk-movements signals?
5. The parameters HRV-HF and HRV-VLF resulted to be re-arranged in the post-intervention. Although the re-arrangement of HRV-VLF is extensively discussed, the same does not happen for HRV-HF. More discussion on the re-arrangement of this parameter would be helpful in having a better overview of the study results.
Minor revisions:
Line 27: the acronym (SNS) lacks its original full name "Sympathetic Nervous System"
Line 28: error in "(SFS)" instead of "(SNS)"
Line 71: unclear the meaning of "sensorized protocol", reformulate?
Line 87: error in sit. Correct in "sat" to maintain syntactic coherence.
Line 131: error in "cardiac:variables:"
Line 135-6: repetition "but it was based on a parallel analysis based on the parallel analysis on the scree-plot".
143: repetition "ARIANNA is ARIANNA is".
145: remove "the" from “architecture of the ARIANNA".
159: substitute the term "changed" instead of "change”.
Affiliation numbers do not match with entitites.
Comments on the Quality of English LanguageI suggest proofreading and english editing perhaps by a service or a mother tongue.
Author Response
AUTHORS: We would like to thank the Editor and the Reviewers for their general positive comments about our work, for the possibility to resubmit it and for the qualified comments that helped us into improving our manuscript in its revised version.
In the following our step-by-step answers to the comments of the Reviewer 2. We have also reported here the new/changed parts of the revised manuscript between apices .
Reviewer 2
General comment:
In line with the 4E cognition theory, the study is carried out with the attempt to apply a new integrative approach called "embodimetrics" to the study of cognitive constructs. Here the performance at the stroop task - one of the literature's gold standard for attentional measures - was measured combining kinestetic and physiological variables. The originality of the study relied on the inclusion of artificial neural networks analyses to further confirm the predictability results from the statistical analyses. The use of a multimodal approach, both theoretical and technological, helps optimize and improve the study of psychological complex phenomena.
AUTHORS: We would like to thank the reviewer for his/her positive comments about our work and for the following suggestions that helped us to improve our manuscript in its revised version.
Main revisions:
R2: 1. The gender differences reported in the previous literature brought the authors to focus on a full women sample. The reason behind this choice (if there is any) is not clear or reported.
AUTHORS: According to this comment we have now specified the reasons behind this choice in the following sentence in which we cited 3 important references:
“Given previous findings indicating gender differences in autonomic cardiac control [17], trunk accelerations [18], and Stroop task performance [19] we enrolled only women in this study to simplify the variables to be controlled.”
- Botek, M., Krejčí, J., & McKune, A. Sex Differences in Autonomic Cardiac Control and Oxygen Saturation Re-sponse to Short-Term Normobaric Hypoxia and Following Recovery: Effect of Aerobic Fitness. Frontiers in endo-crinology, 2018, 9, 697.
- Mazzà, C.; Iosa, M.; Picerno, P.; Cappozzo, A. Gender differences in the control of the upper body accelerations during level walking. Gait Posture, 2009, 29(2), 300-303.
- Mekarski, J.E.; Cutmore, T.R.; Suboski, W. Gender differences during processing of the Stroop task. Percept Mot Skills. 1996, 83(2), 563-8.
R2: 2. For the second phase of the assessment, participants had to follow a protocol consisting of activities such as walking and balance exercises. The study reports a general adherence from participants which is vague and lacks control measures. Devices such as smart-watches or fitness-watches could be given to participants as to better track their activities, and most importantly provide quantitative measures to statistically analyse. Overall, this would confer more reliability to this second phase of the experimental protocol.
AUTHORS: The referee is right, so we have added the following sentence about further studies in the paragraph about the limits of our study:
“Then, some variables were only qualitatively assessed: further researchers on this topic should be conducted in controlled environments with quantitative monitoring of participants’ adherence to the protocol, for example using smartwatches to record the frequency and intensity of physical activities.”
R2: 3. The outline of the independent and dependent variables used in the study are not enough clear, as well as the hypotheses. This may be improved by creating sub-sections in the protocol paragraph. In a first section the variables could be described in their abbreviations and functioning, and separately in another section they could be explained in their expectations/hypotheses.
AUTHORS: To satisfy this request and, at the same time, to avoid excessive complications into the manuscript we have simply added the following sentence:
“The input layers (corresponding to the independent variables) were: trunk ROM, trunk Symmetry Index, HRV-VLF, HRV-LF and HRV-HF, whereas two output (de-pendent variables) were predicted: Stroop Accuracy and Stroop NTCT.”
R2: 4. The choice to use the sensory trunk band device "Beyond Inertial" is incomplete to the aim of this study. Is it justified by the advantage of combining both the collection of both heart-physiological and trunk-movements signals?
AUTHORS: As the reviewer said, that was one of the advantage to use this device. We have not better detailed also the other reason adding the following sentence (and also adding the new reference 15):
“It has been reported that an increase in trunk mobility may alter the sympathovagal balance, thereby modifying HRV [15].”
- Farinatti, P. T., Brandão, C., Soares, P. P., & Duarte, A. F. Acute effects of stretching exercise on the heart rate variability in subjects with low flexibility levels. Journal of strength and conditioning research, 2011, 25(6), 1579–1585.
R2: 5. The parameters HRV-HF and HRV-VLF resulted to be re-arranged in the post-intervention. Although the re-arrangement of HRV-VLF is extensively discussed, the same does not happen for HRV-HF. More discussion on the re-arrangement of this parameter would be helpful in having a better overview of the study results.
AUTHORS: We have reformulated the paragraph commenting this aspect as follows in Results section:
“Table 1 shows the parameters measured at baseline and post intervention. Firstly, it is notable that HRV-LF did not significantly change, whereas the change in HRV-HF was significant and mainly related to a variation in HRV-VLF, although also this modification did not achieve the threshold for rejecting the null hypothesis.”
In Discussion the following sentence was modified:
“According to previous literature [30], the balance between HRV-LF and HRV-HF was found to be associated not only with cardiac parameters but also with cognitive ones [5] and, as we suggested, also with kinematic parameters. In healthy subjects, the ability to perform symmetric wide rotations depends not only on kinematic functions but also to the perception of the trunk mid-line [25], afferent feedback and the percep-tion-action link [16].”
Minor revisions:
R2: Line 27: the acronym (SNS) lacks its original full name "Sympathetic Nervous System"
AUTHORS: We have now added the original full name writing: “sympathetic nervous system (SNS)”
R2: Line 28: error in "(SFS)" instead of "(SNS)"
AUTHORS: Thank you, we have corrected the typo.
R2: Line 71: unclear the meaning of "sensorized protocol", reformulate?
AUTHORS: we changes “sensorized protocol” in “protocol utilizing sensors”
R2: Line 87: error in sit. Correct in "sat" to maintain syntactic coherence.
AUTHORS: Thank you, we have corrected the error.
R2: Line 131: error in "cardiac:variables:"
AUTHORS: Thank you, we have corrected the typo.
R2: Line 135-6: repetition "but it was based on a parallel analysis based on the parallel analysis on the scree-plot".
AUTHORS: Thank you, we have deleted the duplicated part.
R2: 143: repetition "ARIANNA is ARIANNA is".
AUTHORS: Thank you, we have deleted the duplicated part.
R2: 145: remove "the" from “architecture of the ARIANNA".
AUTHORS: Thank you, we have deleted “the”
R2: 159: substitute the term "changed" instead of "change”.
AUTHORS: Thank you, we have corrected the typo.
R2: Affiliation numbers do not match with entitites.
AUTHORS: Thank you, we have corrected the affiliations
R2: Comments on the Quality of English Language - I suggest proofreading and english editing perhaps by a service or a mother tongue.
AUTHORS: An English Editing has been performed throughout the whole manuscript.

Round 2
Reviewer 1 Report
Comments and Suggestions for Authors
The reviewer thank the authors for their detailed answers and revision. The authors have successfully addressed the issues raised in the initial review. These revisions have strengthened the manuscript and resolved any major concerns raised in the first revision.
Based on the authors' revisions and the manuscript's overall quality, I recommend accepting it for publication.
Comments on the Quality of English LanguageThe authors have corrected some errors and have improved the quality of English in the manuscript. In general, the English can be improved further. However, more importantly, reviewers now believe that the message that authors want to convey is discernable.